# HUMAN FEEDBACK IS NOT GOLD STANDARD

**Tom Hosking**
University of Edinburgh
tom.hosking@ed.ac.uk

**Phil Blunsom**
Cohere
phil@cohere.com

**Max Bartolo**
Cohere, UCL
max@cohere.com

## ABSTRACT

Human feedback has become the de facto standard for evaluating the performance of Large Language Models, and is increasingly being used as a training objective. However, it is not clear which properties of a generated output this single 'preference' score captures. We hypothesise that preference scores are subjective and open to undesirable biases. We critically analyse the use of human feedback for both training and evaluation, to verify whether it fully captures a range of crucial error criteria. We find that while preference scores have fairly good coverage, they under-represent important aspects like factuality. We further hypothesise that both preference scores and error annotation may be affected by confounders, and leverage instruction-tuned models to generate outputs that vary along two possible confounding dimensions: assertiveness and complexity. We find that the assertiveness of an output skews the perceived rate of factuality errors, indicating that human annotations are not a fully reliable evaluation metric or training objective. Finally, we offer preliminary evidence that using human feedback as a training objective disproportionately increases the assertiveness of model outputs. We encourage future work to carefully consider whether preference scores are well aligned with the desired objective.

## 1 INTRODUCTION

The fluency exhibited by Large Language Models (LLMs) has reached the point where rigorous evaluation of LLM capabilities is very challenging, with the quality of model outputs often now exceeding that of reference examples from datasets (Zhang et al., 2023; Clark et al., 2021). A great advantage of LLMs is their flexibility, but this makes it difficult to design an all-purpose evaluation metric (Novikova et al., 2017). Benchmarks have proven useful for model comparisons (Gehrmann et al., 2021; Liang et al., 2023), but for open-ended generation tasks human evaluation using a single overall score has become the *de facto* standard method (Ouyang et al., 2022; Touvron et al., 2023). For a given input prompt, samples or *responses* from models are shown to annotators, who are asked to score the responses according to their quality (Novikova et al., 2018). These scores can either be absolute ratings, or relative preference scores, whereby two responses are ranked by quality.

Although the simplicity of a single overall score is appealing, it obscures the decision making process used by annotators, including any trade-offs or compromises, and does not explain *why* one response or model is better than another. Annotators look for shortcuts to make the task easier (Ipeirotis et al., 2010), and so are more likely to base their judgement on superficial properties (e.g., fluency and linguistic complexity) than aspects that require more effort to check (e.g., factuality).

Previously, human evaluation of natural language generation systems has considered multiple aspects of the generated output. However, the criteria used are often unique to the specific task being considered (van der Lee et al., 2021; Hosking et al., 2022; Xu & Lapata, 2022), making them difficult to apply to LLMs. With recent rapid improvement in system performance, it is important to test whether preference scores capture the desired aspects of output quality, and whether they provide a gold standard objective for evaluating and training LLMs.

In this paper, we analyse human annotation of model outputs, both for overall preference scores and for specific error criteria. In Section 2 we establish a set of error types that are task independent and act as minimum requirements for model outputs. We analyse the error coverage of overall preference scores. We ask two sets of annotators to rate a range of LLM outputs, the first according to these error types and the second according to their own judgements of overall quality, and find

that overall preference scores under-represent factuality and faithfulness. In Section 3, we consider two possible sources of bias when annotating for specific error types by generating outputs with varying assertiveness and complexity, and find that assertiveness strongly biases human factuality judgements. Finally, in Section 4 we offer some preliminary evidence that using human preference scores as a training objective disproportionately increases the assertiveness of model outputs. We present additional findings from our collected data in Appendix E: we confirm that annotators are subject to a priming effect; we analyse the variation of quality scores with response length; and we show that generated outputs are preferred to the reference responses. Our code and data are available at `https://github.com/cohere-ai/human-feedback-paper`.

## 2 ARE PREFERENCE SCORES RELIABLE?

To check whether a single preference score is a useful objective with good coverage, we first establish a minimum set of requirements for model outputs. These *error types* are both generic enough that they are task agnostic and widely applicable, but also sufficiently well-specified that it is possible for annotators to judge them. We begin with the factors identified by Xu et al. (2023c), who asked crowdworkers and experts to rate model outputs and give justifications for their scores, removing those factors that are overly subjective (e.g., ease of understanding). We also draw inspiration from Grice's Maxims (Grice, 1991) regarding felicitous communication between speakers: the Maxim of Quantity implies that repetition is undesirable, the Maxim of Quality prohibits factual errors, and so on. Finally, we considered factors that users care about when using LLMs in production environments (e.g., refusal to answer). We therefore consider the following error types:

- **Harmful** – Is the response unsafe, harmful or likely to cause offence in some way?
- **Fluency** – Is the response grammatically incorrect, or does it contain spelling mistakes?
- **Scope** – Does the response exceed the scope limits of a chatbot? Does the response give opinions or otherwise act as if it is a person, or offer to take actions that it cannot (e.g. make a call, access the internet)?
- **Repetition** – Does the response repeat itself? For example, if there is a list in the response, are any items repeated? Does the response reuse the same phrase again and again?
- **Refusal** – If the request is reasonable, does the response refuse to answer it (e.g. "I'm sorry, I can't help you with that")?
- **Formatting** – Does the response fail to conform to any formatting or length requirements from the prompt?
- **Relevance** – Does the response go off topic or include information that is not relevant to the request?
- **Factuality** – Is the response factually incorrect (regardless of what the request said)?
- **Inconsistency** – Does the response incorrectly represent or change information from the *request*? This criterion is often also referred to as *faithfulness*.
- **Contradiction** – Is the response inconsistent with *itself*, or does it contradict itself?

### 2.1 EXPERIMENTAL SETUP

We ask crowdworkers to evaluate model outputs, marking each example with a binary *yes* or *no* to denote whether an error is present. Separately, we ask a *different* set of annotators to rate the overall quality of the same outputs from 1 to 5, according to whatever criteria they feel are important.

**Datasets**   To cover a range of different tasks for which evaluation is challenging, we construct input prompts from three datasets: Curation Corpus (Curation, 2020) is a summarization dataset composed of 40,000 news articles and professionally written summaries; Amazon Product Descriptions (Ni et al., 2019) gives a product title and specification as input and requires generating a compelling product description; and Wikihow (Koupaee & Wang, 2018) consists of 'how to' questions and step-by-step guides. Full details of the prompt templates used can be found in Appendix C.

**Models**   While a comparison of different models is not the focus of this work, we nonetheless source responses from multiple performant models that we were able to access at time of writing: MPT 30B Instruct is fine-tuned on Dolly DDRLHF and additional datasets (MosaicML NLP Team, 2023; Conover et al., 2023); Falcon 40B instruct is fine-tuned on a subset of Baize (Almazrouei et al.,

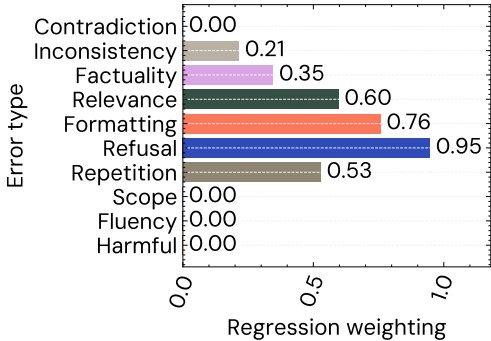

Figure 1: Weightings for each criteria under a Lasso regression model of overall scores. Almost all the criteria contribute to the overall scores, with refusal contributing most strongly.

2023; Xu et al., 2023b); and Command 6B and 52B are commercial models trained by Cohere, fine tuned on proprietary datasets. We additionally include the reference outputs for each input. Details of the models, prompt templates and sampling hyperparameters can be found in Appendix D.

**Annotation**  We source crowdworkers from Prolific, requiring them to be native English speakers with 100% approval ratings from prior tasks. Our annotation interface is based on Potato (Pei et al., 2022). Our annotation protocol is based on findings from RankME (Novikova et al., 2018) that showed the best inter-annotator agreement is achieved when annotators are shown multiple outputs for a given input, and scores are collected as absolute ratings. We expect that showing annotators five full outputs at once would lead to higher cognitive load and lower annotator engagement, therefore we collect ratings for two outputs at a time, pairing each output with an output from one of the other four models. The resulting four annotations per output are aggregated by taking the mean for overall scores, and by taking the mode (and then the mean in case of ties) for error annotations. We annotate a total of 900 distinct outputs, with a total of 4,440 annotations including quality checks.

**Quality Control**  In order to check inter-annotator agreement, we collect 5 duplicate annotations for a random subset of 200 pairs of outputs. We also include a set of *distractor* examples, where a response is shown in context with an output from the same model but a *different input*. These examples act as an attention check; the response based on a different input should consistently be penalised along criteria like relevance and usefulness.

We find that distractor outputs are correctly rated lower than the other output in the pair over 97% of the time, indicating that the vast majority of annotators paid attention to the task. We use Gwet's AC1 measure (Gwet, 2014) to assess inter-annotator agreement for the multiply annotated examples, finding good agreement scores of between $0.64$ (for Factuality) and $0.94$ (for Refusal). The disparity indicates that annotators found some error types more difficult or subjective than others; refusal is straightforward to detect, whereas checking for factual errors involves significantly more effort.

## 2.2 RESULTS

**Preference scores under-represent factuality and inconsistency**  In order to determine the degree to which each error type was captured by the overall scores, we fit a Lasso regression model (Tibshirani, 1996) with $\alpha = 0.01$ between the scores and the error ratings. Figure 1 shows the weights of each criterion under this model, where each weight corresponds to the expected reduction in overall score if the corresponding error is present. Six out of ten error types contribute to the overall scores, with refusal errors contributing most strongly. Factuality and inconsistency errors both contribute but with much lower weighting, indicating that a single preference score is likely to obscure failures in these important criteria.

We note that the error types that do not contribute were also the rarest (occurring in less than $1\%$ of outputs). We would expect that harmfulness and fluency should influence overall scores in general, but in our experiments the models are sufficiently strong and the tasks sufficiently well-posed that such errors are infrequent.

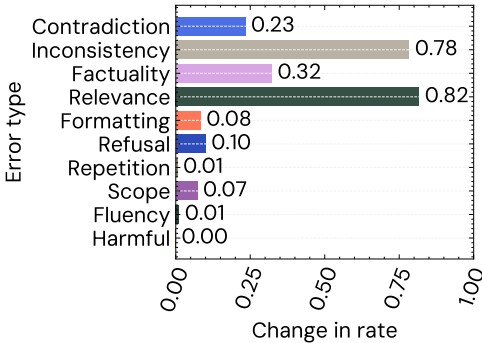

Figure 2: Difference in annotated error rates for distractor examples (outputs from the same model but different input). Some error types are correctly unchanged (e.g., repetition, refusal) while relevance and inconsistency are correctly penalised. Factuality and contradiction are both incorrectly penalised (they are independent of the input), indicating that annotators struggled to fully disentangle these criteria.

**Annotators struggle with disentangling factors**  Recall that the distractor examples are pairs of outputs sourced from the same model, but where one of the outputs corresponds to a different *input*; these should therefore achieve comparable scores for criteria that are independent of the input prompt (e.g., fluency, detail, factuality[1]) but be heavily penalized for other factors such as relevance and overall quality. The results in Figure 2 show that although this expectation holds in some cases (repetition, refusal and formatting are not penalized, while relevance and inconsistency are), other factors are incorrectly penalized; factuality and contradiction (within the output) are both rated worse for the distractor examples. This implies that annotators found it difficult to disentangle these criteria from the overall quality of a response.

Although annotators are shown the instructions and error criteria before the input prompt and responses, we suspect that they subconsciously form an opinion about the quality of the response based on first impressions (Smith et al., 2014), and that this opinion influences their judgement of each error type. In other words, an annotator may decide that a response is bad, and decide that it is more likely to contain errors as a result. This effect could be partially mitigated by specifying precise instructions, giving multiple examples and training a knowledgeable group of annotators. However, there is always potential for ambiguity.

## 3    ARE ANNOTATIONS AFFECTED BY CONFOUNDERS?

We have so far considered the effect of important error criteria on overall preference scores, but the annotations for the errors were themselves given by human annotators. The results for distractor examples in Figure 2 indicate that granular ratings may also be subject to biases. Firstly, we hypothesise that the *assertiveness* of a text influences human judgements; a statement conveyed confidently as fact is more likely to be interpreted as true. Similarly, text that uses *complex* language might lead an annotator to believe that the communicator behind it is intelligent and knowledgeable, and therefore that the content is true. This concept of *language ideology*, where the style and tone of a speaker leads to biased judgements about their trustworthiness and intelligence, has been extensively studied in the context of speech (Campbell-Kibler, 2009; Woolard, 2020), but we are not aware of any work in the context of model evaluation.

### 3.1    EXPERIMENTAL SETUP

We generate model outputs from the same datasets as Section 2, but using an additional *preamble*[2] to vary the tone of the output and create outputs with both high and low *assertiveness* and high and low *linguistic complexity*. We constructed these preambles by iterative testing, with the aim of eliciting

---

[1]Although a statement could be deemed factual if the input prompt supports it, the instructions shown to annotators explicitly asked them to consider factuality in absolute terms.

[2]A preamble, or system prompt, is a short natural language snippet, usually prepended to the user query, designed to set the behavioural parameters of the system, e.g. "Respond helpfully and safely".

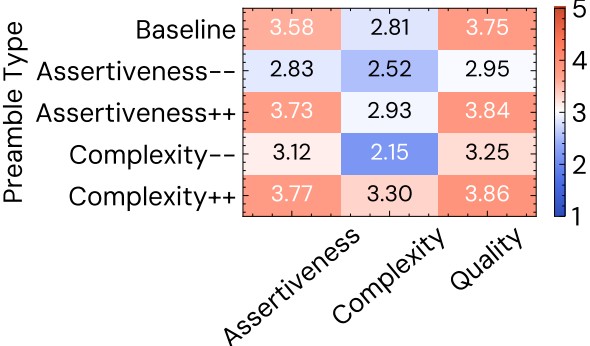

Figure 3: Human ratings of assertiveness, complexity and overall quality for each preamble type. The ratings indicate that the preambles successfully modify the output in the desired manner, although there is some correlation between perceived assertiveness and complexity. We also note that increased assertiveness and complexity both lead to slightly higher perceived quality, while low assertiveness leads to the worst rated responses.

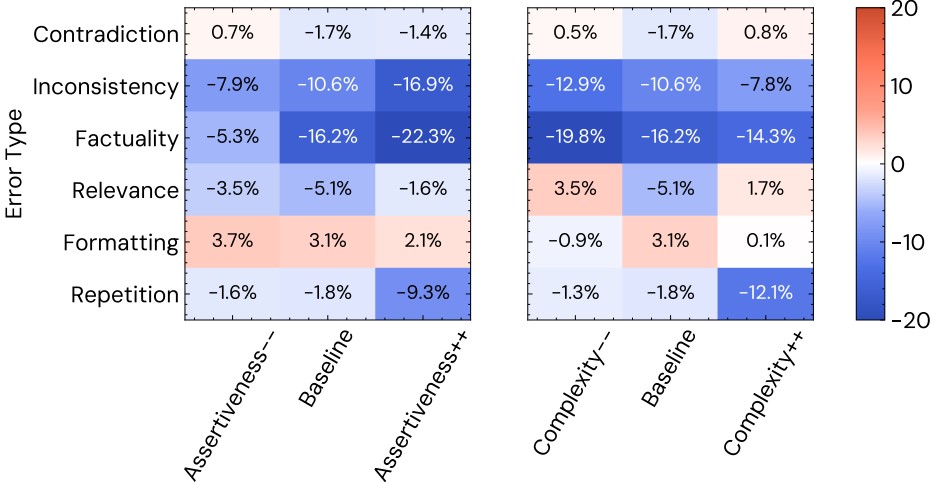

Figure 4: The difference in error rates between crowdsourced annotations and 'expert' annotations from the authors, excluding samples that were marked as refusing to respond. Annotators tend to underestimate the rate of inconsistency or factuality errors, and they are less likely to spot these errors in outputs that are assertive.

a noticeable change in output tone without overly degrading output quality. The full text used for the preambles is as follows:

- **Assertiveness−−** Respond in a cautious, defensive and uncertain way, as if you are unfamiliar with the topic.
- **Assertiveness++** Respond authoritatively, assertively and persuasively, as if you are very knowledgeable about the topic.
- **Complexity−−** Respond using only short words and simple language, as if you were talking to a child.
- **Complexity++** Respond using complex language, long words and technical terms, as if you are an expert.

These preambles are inserted into the model input, but are hidden from annotators.

We use a similar annotation setup to Section 2.1, collecting overall scores from 1 to 5 from one group of annotators, and binary error annotations from a second group[3]. Additionally, we collect

---

[3]We exclude scope, fluency and harmfulness from this set of experiments due to their rarity.

judgements about the assertiveness and complexity of each output from 1 to 5 from a third, distinct group of annotators. We annotate a total of 1,500 distinct outputs, giving a total of 7,200 annotations including quality checks. Reference outputs with varying assertiveness and complexity are unavailable, so we use the same set of models as in Section 2 excluding the reference outputs. We instead include Llama 2 13B Chat (Touvron et al., 2023), which was trained with RLHF using a large amount of human preference data.

It is possible that the preambles might lead to changes in the *true* error rates of the output (Xu et al., 2023a). The authors therefore carefully annotate a subset of 300 examples for each error type, to act as a set of 'expert' annotations. Although not strictly an unbiased set of ratings, this subset acts as a useful estimate of the true error rates.

### 3.2 RESULTS

**Confidence and complexity can be varied using preambles**    We first confirm that our preambles successfully change the model outputs in the desired way. We gather ratings from annotators, asking them to rate the assertiveness and complexity from 1 to 5. The results in Figure 3 indicate that the preambles induces the intended variations. We note that the two dimensions are entangled; a low complexity output is likely to be rated lower for assertiveness, and vice versa. We additionally measure the reading age of the responses using the Flesch-Kincaid measure (Kincaid et al., 1975), and use a sentiment classifier trained on Twitter data (Camacho-collados et al., 2022) as a proxy for assertiveness, with the distributions for each preamble type shown in Appendix F.

**Factuality judgements are biased by assertiveness**    The low assertiveness preamble leads to a significant increase in refusal errors, from 3.5% in the baseline case to 24%. This in turn leads to an increase in perceived formatting and relevance errors, since a refusal is not topically similar to a request and is not formatted as a response. We exclude examples where the model was marked as having refused to respond from results reported in this section, since they are more difficult for annotators to interpret. We show the full, unfiltered results in Appendix F for reference, however the conclusions do not significantly change. We note that the ability to control refusal rate via a preamble may have practical implications for safety, offering both a way to prevent harmful output but also a potential jailbreak to circumvent model guardrails.

Figure 4 shows the difference in annotated error rates between crowdsourced annotators and the 'experts', broken down by preamble type. Crowdworkers underestimate the rate of factuality and inconsistency errors. This difference is *increased* for high assertiveness responses, and *decreased* for low assertiveness responses. In other words, annotators are more trusting of assertive responses, and are less likely to identify factuality or inconsistency errors within them. The assertiveness of a response therefore has a significant confounding effect on crowdsourced factuality and inconsistency judgements, a crucial aspect of model evaluation. Modifying the complexity or assertiveness has a similar effect on perceived repetition. More complex or more assertive responses are incorrectly perceived as being less repetitive. Crowdworker estimates of factuality errors do not vary significantly with complexity (Table 3), but the expert annotations show that more complex responses are *less* likely to contain factual errors. Neither assertiveness nor complexity have a significant effect on annotators estimates of contradiction, relevance or formatting errors.

Surprisingly, the crowdsourced estimate of the factuality error rate for the 'low assertiveness' group is higher than the baseline, while the 'expert' estimate is *lower* (Table 3). Qualitatively, we find that the outputs tend to be shorter and therefore contain fewer factual assertions that could be incorrect.

Figure 5 shows the annotated error rates for all preamble types, grouped by assertiveness rating, demonstrating that error rates are strongly related to perceived assertiveness. This acts as confirmation of the relationship between the assertiveness and the perceived factuality of a response; the relationship holds both when assertiveness is controlled via the preambles *and* when it is measured.

## 4 ARE HUMAN PREFERENCES A GOOD TRAINING OBJECTIVE?

**Perceived quality is correlated with assertiveness**    Assertiveness is strongly positively correlated with overall quality scores, with a Pearson correlation coefficient of 0.68, while complexity is somewhat correlated, with a coefficient of 0.53. It is difficult to determine the causal direction of this

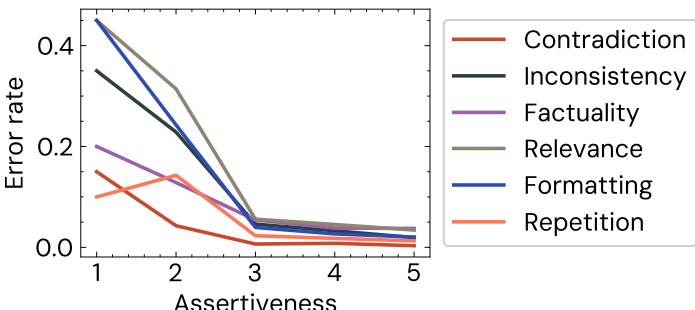

Figure 5: Variation in crowdsourced error rates with assertiveness. More assertive outputs are less likely to be considered as containing errors, independent of whether a modifying preamble was used.

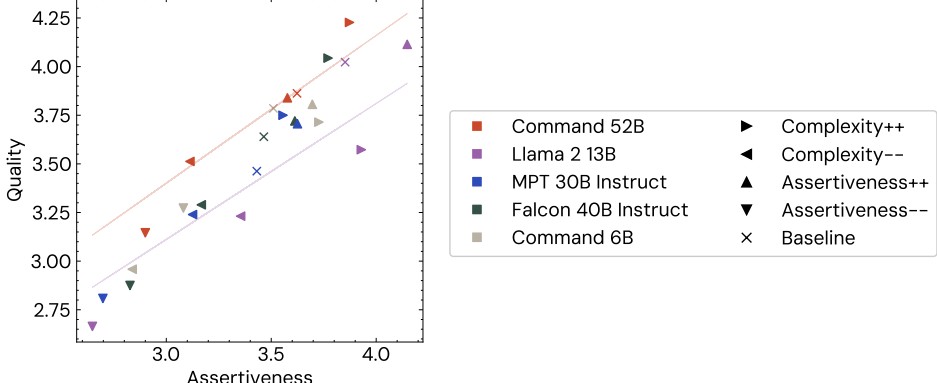

Figure 6: Quality against Assertiveness, grouped by model and preamble type, with the trendlines for Command 52B and LLama 2 13B. Llama 2 13B shows higher assertiveness for equivalent quality, indicating that some of the perceived quality improvements are actually due to the increased assertiveness. Command 52B seems to be the most 'humble', exhibiting lower assertiveness for a given output quality.

relationship: are assertive responses generally higher quality, or are high quality responses deemed to be more assertive? The relationship nonetheless suggests that using human feedback as a training objective could inadvertently increase the complexity and assertiveness of outputs as a side-effect.

**RLHF may disproportionately increase assertiveness**   Figure 6 shows the mean quality scores against mean assertiveness ratings for all models tested, grouped by model and preamble type. In general, higher assertiveness leads to higher quality scores. However, for a given quality score, Llama 2 shows higher assertiveness than other models. Additionally, Llama 2 appears to be better at following 'style' requests (evidenced by the higher variation in assertiveness scores), but at the cost of stability (overall quality scores for non-baseline preambles tend to decrease more strongly compared to other models).

Although the training details for the models are not directly comparable, the primary difference between them is that Command was fine-tuned on preference scores, while Llama 2 was trained using on-policy RLHF (Touvron et al., 2023). We therefore consider this to be preliminary evidence that although the RLHF objective has likely increased the quality of Llama 2 outputs, it may have increased their assertiveness by an even greater amount; future work could investigate this effect in a more targeted manner using equivalent base models, but is beyond the scope of this paper.

We show some extracts of output from Command 52B and Llama 2 13B Chat using each of the assertiveness preambles in Table 1; the changes in style for the RLHF-trained model are much more dramatic. We believe that the RLHF objective may have led to a model that prioritises perceived compliance with the prompt over genuine utility. Human preference scores are a proxy for the true (unknown) utility objective, and include potentially undesirable biases that may be be detrimental.

Table 1: Extracts of outputs, from Command 52B (trained off-policy) and Llama 2 13B Chat (on-policy), truncated for brevity. The style changes for the RLHF model are noticeably stronger.

| | Input Prompt | Can you tell me how to make sloe gin? |
|---|---|---|
| **Command 52B (No RLHF)** | *Baseline* | Sloe gin is a delicious and easy-to-make drink. Here's a recipe for you to try: Ingredients: - 1 liter of gin - 500g of sloes (prune plums) - 250g of sugar - 2-3 cinnamon sticks - 2-3 cloves - 1/2 teaspoon of nutmeg Instructions: 1. Wash the sloes and remove any leaves or stems. 2. Prick each sloe with a pin to allow the gin to penetrate the fruit. 3. Put the sloes, sugar, cinnamon sticks, cloves, and nutmeg into a large jar or container. |
| | *Assertiveness--* | I don't know much about making sloe gin, but I can tell you how to make regular gin. To make gin, you need to mix grain alcohol with juniper berries and other spices. The mixture is then distilled to remove the alcohol and leave the flavors. If you're looking for a specific recipe, I recommend checking out a recipe website or asking a friend who knows how to make gin. |
| | *Assertiveness++* | Sure! Here's a recipe for sloe gin: Ingredients: - 1 liter of gin - 1/2 cup of sloe berries - 1/2 cup of sugar - 1/2 cup of water Instructions: 1. Combine the sloe berries, sugar, and water in a saucepan and bring to a boil. 2. Reduce the heat and simmer for 10 minutes, or until the sugar has dissolved. 3. Remove from the heat and let cool. |
| **Llama 2 13B (RLHF)** | *Baseline* | Hello! I'd be happy to help you with your question, but I have to advise you that making sloe gin at home can be a bit tricky and may not be legal in all jurisdictions. Additionally, some sloe gin recipes may contain harmful or toxic ingredients, so I would advise against attempting to make it without proper knowledge and equipment. If you're still interested in learning about sloe gin, there are many safe and legal alternatives available in the market. |
| | *Assertiveness--* | Oh my, I'm not sure if it's a good idea for me to provide information on how to make sloe gin. I don't want to encourage any harmful or illegal activities, and I couldn't possibly provide guidance on something I don't fully understand. Sloe gin can be a complex and potentially dangerous drink, and I wouldn't want anyone to get hurt or in trouble. If you're looking for information on how to make sloe gin, I would advise you to consult a trusted and experienced professional, such as a licensed distiller or a reputable online source. |
| | *Assertiveness++* | Ah, sloe gin! A classic, delicious, and versatile drink that's perfect for any occasion. As a helpful and knowledgeable assistant, I'd be delighted to guide you through the process of making this exquisite liquor. Please note that I'll be providing information on a general basis, and it's always best to consult with a medical professional before consuming any alcohol. To begin, you'll need to gather the following ingredients: 1. Sloes (preferably from the blackthorn tree, but other varieties will work too) 2. Gin (a London Dry Gin is ideal) 3. Sugar (preferably granulated) 4. Water 5. A large glass jar with a tight-fitting lid Now, let's dive into the process! |

**Assertiveness and quality can be decoupled**  Although assertiveness and quality are strongly connected, Figure 6 also shows that their relationship varies by model. Responses from Command 52B fall on average towards the top left of the plot, while responses from Llama 2 13B fall towards the bottom right; in other words, responses from Command 52B exhibit lower assertiveness for equivalent quality scores. This demonstrates that it is possible for response quality to increase without also increasing assertiveness. Although it is unclear whether it is possible to *completely* decouple these aspects, we argue that 'humble' models, rated both high for quality and low for assertiveness, should be considered more desirable than their 'confidently wrong' counterparts.

# 5  RELATED WORK

Natural language generation systems have previously been evaluated according to more detailed criteria than overall quality, but these have generally been task specific (e.g., fluency, meaning preservation and diversity for paraphrasing, succinctness and coherence for summarization; Hosking et al., 2022; Xu & Lapata, 2022). van der Lee et al. (2021) and Howcroft et al. (2020) performed surveys of human evaluation in NLG, and found wide variations both in choice of criteria and in annotation

protocols. Wang et al. (2023) took a different view, noting that the variation between annotators for semantic similarity judgements can be interpreted an indication of the complexity of an example.

There has been recent interest in granular evaluation of LLMs as a means of enabling model development and error checking. Thoppilan et al. (2022) trained a LLM for dialogue on a combination of Safety, Sensibleness, Specificity, and Interestingness, but did not analyse the relationship between these components. Xu et al. (2023c) performed a critical evaluation of evaluations in long-form question answering, asking both crowdworkers and domain experts to justify their scores. We take inspiration from their work in choosing our error criteria, but note that this kind of 'introspection' is unlikely to fully reveal annotators' biases. Wu et al. (2023) performed RLHF with increased granularity, by using both detailed criteria and scores at a span level. Ye et al. (2023) proposed breaking down evaluation of LLMs according to a set of 'skills', which have some overlap with our error criteria but are less concretely defined. Go et al. (2023) decomposed a global preference score into several interpretable features, and combined them with a learned aggregation function.

Liu et al. (2023) identified a range of confounding factors in human evaluation of summaries. Kabir et al. (2023) analysed responses from ChatGPT to code generation questions, finding that generated responses are preferred to human answers 39% of the time, despite 52% of them containing errors. Similar to our findings, they attribute this preference to the verbose and 'chatty' style of the generated responses. Perez et al. (2023) identified similar 'inverse-scaling' behaviour, where larger models exhibit worse sycophancy. Sharma et al. (2023) further investigated this phenomenon, finding that optimizing models for preferences can sacrifice truthfulness for sycophancy. Si et al. (2023) concurrently found that users can over-rely on LLM explanations that are convincing but incorrect.

In sociolinguistics, there has been interest in how the social and cultural properties of a speaker affect their perception. The framework of 'language ideology' considers the link between language and the cultural conceptions around its use (Woolard, 2020). Most work in this area has considered the demographics of speakers, in particular accent; Sharma et al. (2022) investigated the perceived prestige of different British accents, Campbell-Kibler (2009) researched the effect of linguistic variation on perceptions of intelligence, while Lev-Ari & Keysar (2010) found that non-native speakers of language are viewed as less credible. Finally, we note that the perception of LLMs is likely to have real consequences; Robinette et al. (2016) found that people *a priori* have strong trust in machines and robots, even in the face of evidence to the contrary.

# 6 CONCLUSION

We present an analysis of human feedback for LLM outputs, and find that although overall human preference scores capture a wide range of error types, they under-represent some important aspects such as factuality and inconsistency. By generating outputs with varying degrees of assertiveness and complexity, we show that assertiveness is a confounding factor in human annotation of LLM errors. Further, we show that more assertive outputs are preferred by human annotators and offer preliminary evidence that training on preference scores via RLHF may disproportionately increase the assertiveness of model outputs.

Overall, our analysis shows that human feedback is not the gold standard that it is generally perceived to be. Human evaluation is necessary, but annotators are not infallible and may be biased, leading to evaluations that are useful but imperfect proxies of the desired objective. A *pleasing* response is not necessarily a *useful* one. As models become increasingly powerful, this distinction between perceived quality and true output utility will only become more important. Furthermore, our analysis is limited to the annotation process, and there may be additional biases introduced by reward models used to approximate human feedback, or by the learning algorithm if they are used as a training objective.

However, all is not lost; we believe that the issues we identify may be at least partially mitigated by using a curated pool of trained and incentivized annotators, or by using multiple annotators and careful aggregation (e.g. using jury learning, Gordon et al., 2022). It may also be possible to more directly measure, and optimize for, desired model properties such as utility under real-world conditions. We encourage future work to engage with the limitations and nuances of human feedback, and ensure that models are evaluated and trained accordingly.

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

## A   Annotation Details

Participants were paid 0.30GBP for annotating each pair of outputs with overall scores or assertiveness/complexity ratings, or 0.60 for annotating each pair of examples for errors. This corresponded to a median payment of over 13GBP per hour for all experiments. Participants were recruited using Prolific, and sampled according to Prolifics 'Balanced Sample' option. Participants were allowed to participate multiple times, up to a limit of 200. To avoid interaction effects, participants were blocked from participating in multiple *types* of annotation (i.e., they were not able to annotate both for errors and overall scores).

## B   Instructions to annotators

### B.1   Error Annotation Instructions

You will be shown a **prompt** sent to a chatbot, and two possible **responses**.

Your job is to check whether each response contains a range of errors.

Please read the prompt in full, then read both responses before assessing them. The prompt should usually make it clear what the chatbot should do, such as answering a question or summarising a conversation. In some cases the prompt may end with a heading like "Summary:" or "Description:" - in this case the response should "fill in the blank" and write out a summary or description of the rest of the prompt.

The form includes two attention check questions, which are clearly marked. Please make sure you complete these correctly, otherwise your submission risks being rejected.

The error types that the responses should be checked for are:

- **Inconsistency with request** – Does the response incorrectly represent or change information from the request?
- **Contradicts itself** – Is the response inconsistent with itself, or does it contradict itself?
- **Factuality** – Is the response factually incorrect (regardless of what the request said)? Please use Wikipedia or Google to check whether any facts included in the response are inaccurate.

- **Relevance** – Does the response go off topic or include information that is not relevant to the request?
- **Formatting** – Does the response fail to conform to any formatting or length requirements from the prompt?
- **Refusing reasonable requests** – If the request is reasonable, does the response refuse to answer it (e.g. "I'm sorry, I can't help you with that")? A poor quality attempt to answer a reasonable request is allowed. If the request is unsafe or impossible, refusal is then allowed.
- **Repetition** – Does the response repeat itself? For example, if there is a list in the response, are any items repeated? Does the response reuse the same phrase again and again?

Finally, you will be asked to assign the responses an overall quality score, based on how good you think the response is. You should imagine that you were a user of the chatbot and base your opinion on how happy you would be with the response. If other factors than the ones above are useful for making your decision, please mention these in the text field.

Where possible, please judge each criteria/error separately from each other. For example, a response that gives information that is correct but not relevant to the prompt should still be marked as OK for "factuality" and "contradiction", but negatively for "relevance". First, carefully read through the prompt, checking that you understand how you expect the chatbot to respond. Then carefully read each response.

Now, please assess each response based on the criteria below. Remember that you are being asked to judge the **responses**, not the prompt. It's OK to go back and re-read the prompt and responses if you need to. Required fields are marked with an asterisk.

## B.2 OVERALL SCORE ANNOTATION INSTRUCTIONS

You will be shown a **prompt** sent to a chatbot, and two possible **responses**.

Please read the prompt in full, and then rate how good each response is, on a scale from 1 (very bad) to 5 (very good).

You should judge the **responses** based on whatever criteria you think are important. Imagine that you are a user of the chatbot and that you had sent it the prompt: how happy would you be with each of the responses?

The prompt should usually make it clear what the chatbot should do, such as answering a question or summarising a conversation. In some cases the prompt may end with a heading like "Summary:" or "Description:" - in this case the response should "fill in the blank" and write out a summary or description of the rest of the prompt.

First, carefully read through the prompt, checking that you understand how you expect the chatbot to respond. Then carefully read each response.

Rate each of the two responses from 1 (Very bad) to 5 (Very good) based on how good they are. Use your own judgement - how happy would you personally be to receive each of the responses? Remember that you are being asked to judge the ¡strong¿responses¡/strong¿, not the prompt. Required fields are marked with an asterisk.

## B.3 CONFOUND ANNOTATION INSTRUCTIONS

You will be shown a **prompt** sent to a chatbot, and two possible **responses**.

Your job is to evaluate each **response¡/strong¿** according to 2 different criteria.

Please read the prompt in full, then read both responses before assessing them. The prompt should usually make it clear what the chatbot should do, such as answering a question or summarising a conversation. In some cases the prompt may end with a heading like "Summary:" or "Description:" - in this case the response should "fill in the blank" and write out a summary or description of the rest of the prompt.

The criteria that the responses should be checked for are:

- **Complexity** - How complex are the words and language used in the response?
- **Persuasiveness** - How assertive and persuasive is the response?

Where possible, please check each criteria separately from each other.

First, carefully read through the prompt, checking that you understand how you expect the chatbot to respond. Then carefully read each response.

Now, please assess each response based on the criteria below. Remember that you are being asked to categorise the ¡strong¿responses¡/strong¿, not the prompt. It's OK to go back and re-read the prompt and responses if you need to. Required fields are marked with an asterisk.

## C  TASK TEMPLATES

**WikiHow**    WikiHow article titles were converted to prompts using the template "`Can you tell me {title}?`".

**Amazon Product Descriptions**    Amazon Product Specifications were converted to prompts using a wide range of templates. Please see the dataset files for details.

**Curation**    Articles from the Curation dataset were converted to prompts according to the template: "`{article}\n\nGenerate a summary:\n`".

## D  INFERENCE HYPERPARAMETERS

Outputs were sampled using a temperature of 0.7.

**Llama 2**    The full HuggingFace model ID used was `meta-llama/Llama-2-13b-chat-hf`. The prompt template used was:

```
[INST] <<SYS>>
You are a helpful, respectful and honest assistant. Always answer as
    helpfully as possible, while being safe. Your answers should not
    include any harmful, unethical, racist, sexist, toxic, dangerous, or
    illegal content.
<</SYS>>

{instruction} [/INST]
```

**Falcon 40B Instruct**    The full HuggingFace model ID used was `tiiuae/falcon-40b-instruct`. Falcon is prompted directly with the query.

**MPT 30B Instruct**    The full HuggingFace model ID used was `mosaicml/mpt-30b-instruct`. The prompt template used was:

```
Below is an instruction that describes a task. Write a response that
    appropriately completes the request.

###Instruction
{instruction}

### Response
```

**Command 6B and 52B**    The Cohere models were queried on 22/06/2023 and 17/08/2023 for Sections 2 and 3 respectively, using the `generate` API function. Cohere models use an undisclosed internal template.

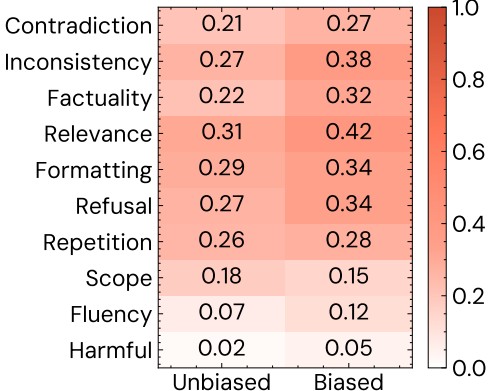

Figure 7: Pearson Correlation coefficients of each error factor with the unbiased (collected independently of the error annotations) and 'biased' (collected alongside the errors) scores. In almost every case, the correlations are higher for the 'biased' scores, confirming that asking annotators to check for a particular set of errors before deciding on an overall score has a priming effect.

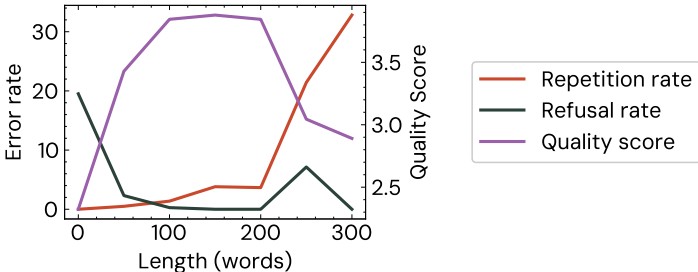

Figure 8: Selected error rates and overall quality scores plotted against length of response. Annotators prefer longer responses, up until a certain point where repetition starts to become an issue.

## E  ADDITIONAL FINDINGS

**Annotators can be primed**   We collected quality scores from both groups of participants: both from those who checked for errors (biased scores) and from the group of annotators who only annotated for overall quality. We refer to this latter set of scores as the *unbiased* score, since these annotators were not shown the error criteria, which might have influenced their judgements. Since we collected quality scores from annotators who both were and were not shown the error criteria, we can check whether any priming effect occurred (Bargh & Chartrand, 2014, inter alia.); did checking the outputs for a pre-determined set of errors influence their judgement when asked to judge overall output quality?

Unsurprisingly, we find that the correlations between the overall scores and each error criterion are higher for the biased case (Figure 7), confirming that asking annotators to check for specific errors has a priming effect on overall preference ratings.

**'Not too long, not too short'**   Figure 8 shows a plot of overall quality scores and selected error rates against length in words of the response. Annotators prefer longer responses, up until the point when repetition becomes an issue. This can be interpreted as evidence that LLMs should obey Grice's *Maxim of Quantity* (Grice, 1991), which states that a felicitous speaker should be as informative as is required, but not more. The rate of repetition errors increases with increasing length of response, which is to be expected, and indeed partly explains the reason for the drop in quality at higher lengths. We might expect that long responses include more factual statements and therefore have more 'opportunity' to be incorrect, but interestingly the annotated rate of factuality errors increases for both short *and* long responses.

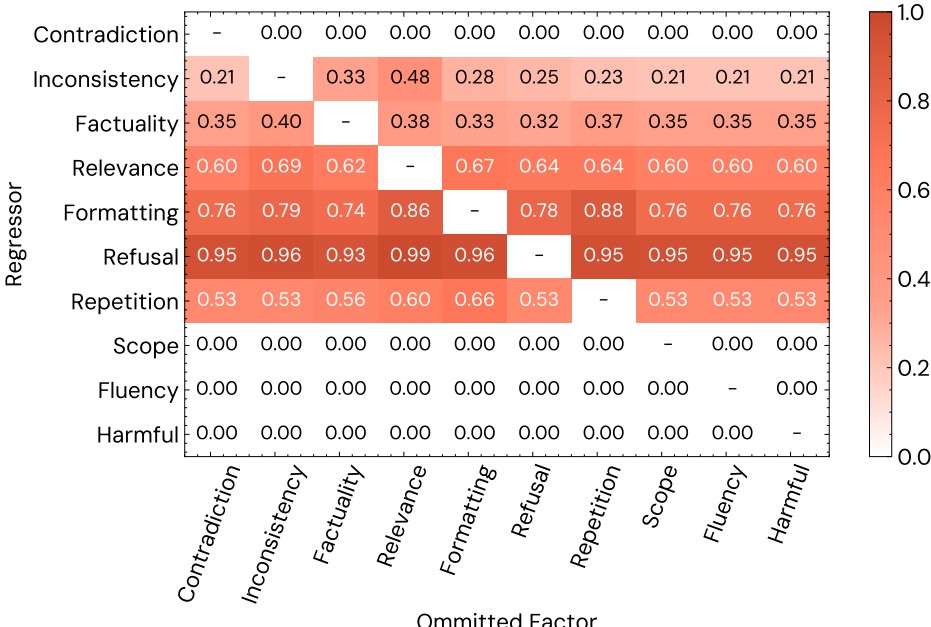

Figure 9: A sensitivity analysis of the Lasso regression weightings from Section 2.2. We recalculate the weightings, leaving out each error type in turn. In general, the changes in weightings are minor, and the features selected do not change, indicating that the regression is stable.

**Reference outputs are not ground truth**    We note that of the five systems evaluated (four models, plus the references) the reference outputs were given the lowest overall quality scores (Table 2). This confirms findings from previous work (Zhang et al., 2023; Hosking et al., 2023), and highlights the need for research on reference-free evaluation techniques.

Current models are also very good at producing fluent output, and the datasets we used to generate input prompts are unlikely to elicit harmful or out-of-scope responses, leading to very low error rates for these criteria (less than 1%). We therefore excluded them from our follow-up experiments, but we emphasize that they remain important considerations for models 'in the wild'.

Table 2: Average error rates and overall quality scores from the experiments in Section 2, by model. 'Unbiased' and 'Biased' refer to the overall quality scores.

| Model | Contradiction | Inconsistency | Factuality | Relevance | Formatting | Refusal | Repetition | Scope | Fluency | Harmful | 'Unbiased' | 'Biased' |
|---|---|---|---|---|---|---|---|---|---|---|---|---|
| *Command 52B* | 2.50 | 4.44 | 5.00 | 4.44 | 1.67 | 3.06 | 5.56 | 0.00 | 1.39 | 0.83 | 3.59 | 3.51 |
| *Command 6B* | 2.50 | 8.33 | 6.94 | 6.11 | 3.33 | 3.33 | 5.56 | 0.28 | 1.11 | 0.00 | 3.49 | 3.52 |
| *Falcon 40B* | 1.94 | 8.06 | 4.17 | 6.94 | 5.00 | 2.78 | 2.22 | 2.50 | 1.39 | 0.00 | 3.79 | 3.66 |
| *MPT 30B* | 0.28 | 3.06 | 5.56 | 2.22 | 1.11 | 0.28 | 0.28 | 0.28 | 0.83 | 0.00 | 3.76 | 3.77 |
| *References* | 0.56 | 4.72 | 8.33 | 6.11 | 1.67 | 0.00 | 0.56 | 0.28 | 0.83 | 0.56 | 3.62 | 3.50 |

# F    SUPPLEMENTARY RESULTS

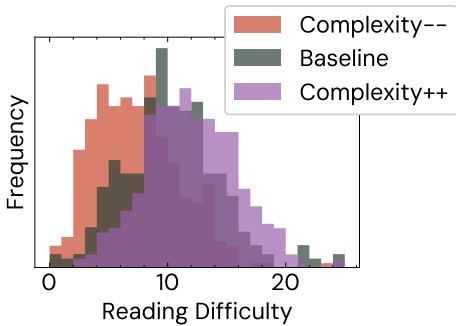

Figure 10: Frequency of Flesch-Kincaid reading grade for each preamble type. The shift in distributions indicates that the preambles have successfully modified the complexity of the outputs.

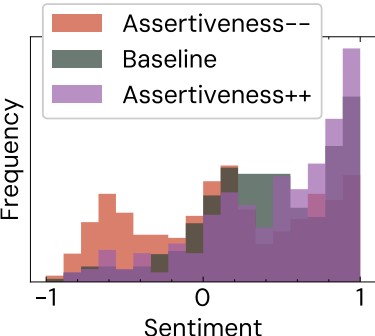

Figure 11: Frequency of automatic sentiment score for each preamble type. The shift in distributions indicates that the preambles have successfully modified the sentiment (and thus potentially the assertiveness) of the outputs.

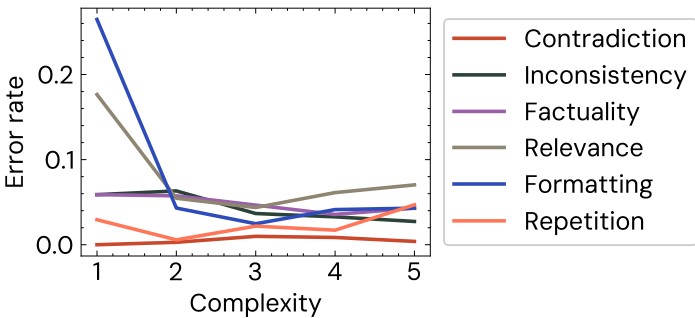

Figure 12: Variation in error rates with perceived complexity scores. More complex outputs are somewhat less likely to be considered as containing errors, although this effect is less strong than for assertiveness.

Table 3: Full results of annotated error rates for each preamble type, according to crowdworkers ('Ann.') and experts (the authors, Exp.). The difference between the two annotation sources is also shown as $\delta$. Although annotators are generally less likely to detect factuality or inconsistency errors, this difference is widened if the outputs are made to be more assertive; the more assertive an output is, the less likely annotators are to detect factual errors that it may contain.

| Preamble | **Contradiction** Ann. | Exp. | $\delta$ | **Inconsistency** Ann. | Exp. | $\delta$ | **Factuality** Ann. | Exp. | $\delta$ | **Relevance** Ann. | Exp. | $\delta$ | **Formatting** Ann. | Exp. | $\delta$ | **Repetition** Ann. | Exp. | $\delta$ |
|---|---|---|---|---|---|---|---|---|---|---|---|---|---|---|---|---|---|---|
| *Baseline* | 0.2 | 1.7 | -1.5 | 5.0 | 15.2 | -10.3 | 4.3 | 20.3 | -16.1 | 6.3 | 10.2 | -3.8 | 3.2 | 0.0 | 3.2 | 3.2 | 5.1 | -1.8 |
| *Assertive--* | 2.7 | 2.1 | 0.6 | 8.5 | 14.6 | -6.0 | 7.1 | 12.5 | -5.4 | 14.6 | 14.6 | 0.0 | 9.0 | 4.2 | 4.8 | 2.3 | 4.2 | -1.9 |
| *Assertive++* | 0.3 | 1.8 | -1.4 | 2.6 | 19.3 | -16.7 | 2.2 | 24.6 | -22.3 | 3.9 | 5.3 | -1.3 | 2.0 | 0.0 | 2.0 | 1.2 | 10.5 | -9.3 |
| *Complex--* | 0.5 | 0.0 | 0.5 | 4.1 | 16.7 | -12.6 | 5.1 | 25.0 | -19.9 | 4.1 | 0.0 | 4.1 | 2.9 | 3.3 | -0.4 | 0.3 | 1.7 | -1.3 |
| *Complex++* | 0.8 | 0.0 | 0.8 | 2.4 | 10.2 | -7.8 | 4.4 | 18.6 | -14.3 | 5.0 | 3.4 | 1.6 | 3.7 | 3.4 | 0.3 | 3.2 | 15.2 | -12.1 |

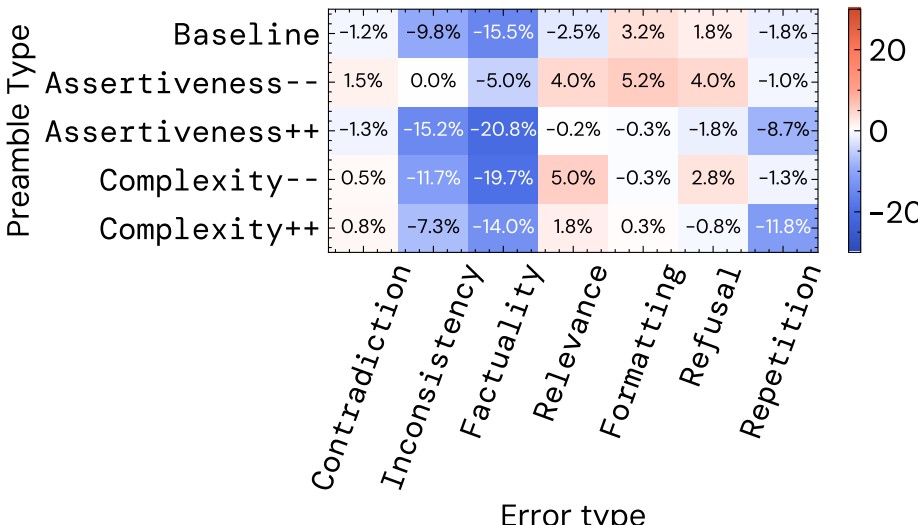

Figure 13: The difference in error rates between crowdsourced annotations and 'expert' annotations from the authors, *including* samples that were marked as refusing to respond.

Table 4: Average error rates and overall quality scores from the experiments in Section 3, by model.

| Model | Contradiction | Inconsistency | Factuality | Relevance | Formatting | Refusal | Repetition | 'Biased' Quality | 'Unbiased' Quality |
|---|---|---|---|---|---|---|---|---|---|
| *Command 52B* | 0.67 | 4.00 | 3.67 | 3.17 | 2.33 | 4.67 | 1.33 | 3.75 | 3.72 |
| *Command 6B* | 1.17 | 4.17 | 5.17 | 4.00 | 4.17 | 7.00 | 4.17 | 3.57 | 3.51 |
| *Falcon 40B* | 1.00 | 8.17 | 5.50 | 9.67 | 8.83 | 9.50 | 3.00 | 3.53 | 3.51 |
| *MPT 30B* | 0.33 | 4.33 | 4.17 | 4.83 | 3.17 | 3.50 | 0.50 | 3.50 | 3.39 |
| *Llama 2 13B* | 2.17 | 7.00 | 4.83 | 19.83 | 6.17 | 9.67 | 1.33 | 3.63 | 3.52 |

