# OpenReview forum: "Human Feedback is not Gold Standard"
_ICLR.cc/2024/Conference — ICLR 2024 poster_

### Official Review · Reviewer_mWEt · 2023-10-31

**Soundness:** 4 excellent
**Presentation:** 3 good
**Contribution:** 3 good
**Rating:** 8
**Confidence:** 3

**Summary:**

The paper is a careful study of biases in human feedback, a critical component of influential approaches to aligning (e.g. RLHF) and evaluating LMs. The authors conduct extensive experiments having humans evaluate responses of several LMs (Cohere models, Mosaic models and LLaMA-2).

First, the authors (i) ask crowdworkers to evaluate LMs’ outputs according to a set of separate criteria as well as give overall score, (ii) train a linear model to predict overall score from scores for separate criteria, (iii) use coefficients of the linear model as a measure of importance of each separate criterion. Model refusing to response is the most important criterion. When the crowdworkers are shown distractor samples (query and response are mismatched), factuality and contradiction (within the output) are both rated worse which implies that annotators fail to disentangle these criteria from the overall quality of a response.

Next, the authors investigate how assertiveness and complexity of LM response affect human judgements. They found that annotators are more trusting of assertive responses, and are less likely to identify factuality or inconsistency errors within them. Furthermore, more complex or more assertive responses are incorrectly perceived as being less repetitive.

Both of these findings are cast a shadow of doubt on whether reward derived from human feedback is the right optimization target for aligning LMs or the right criterion for evaluating them.

**Strengths:**

1. I think the paper addresses a very important and understudied problem, at the very heart of AI safety: how to supervise and evaluated highly capable LMs. The fact that humans are easily misled and prefer convincing-and-assertive-but-subtly-wrong replies over truly correct replies is extremely worrying as it suggests we are training LMs to be deceptive.
2. The paper is well-written and easy to follow.
3. The analyses in the paper are carefully designed and comprehensive. I appreciate that the authors took care of estimating crowdworkers reliability, that their human feedback collection procedure is informed by a body of NLP and linguistics and that they conduct experiments with relatively large models (13B-52B).

**Weaknesses:**

1. One big question not addressed in the paper is how the biases in human judgement scale with human capabilities (e.g. question difficulty, annotator competence) and model capabilities (e.g. model size). Are humans more biased when evaluating responses they’re less competent to evaluate (e.g. medical advice, science explanations)? Are larger models more susceptible to exploit human biases? Overall, I think a good perspective for grounding this paper is scalable oversight [1]: the problem of evaluating agents more capable than the evaluators. LM capabilities will soon surpass human capabilities on many tasks and it’s critical to know whether human feedback will be increasingly misleading. That’s of course a big open problem and I don’t take not addressing it against the paper, it but it would be relatively easy to do experiments with models of the same family but different size or group questions by (perceived) difficulty.
2. I’m not sure I’m convinced that “Human evaluation is necessary” (as stated in the conclusion). Techniques such as RL from AI feedback have been shown to be as effective as RLHF for increasing the harmlessness of LMs [2]. They ultimately rely on humans, but these are human experts involved in prompt engineering and designing rules and constitutions used for generating training data for SFT and training preference models. A big open questions is whether LMs-as-evaluators inherit human biases, even when carefully prompted. A simple experiment addressing this question would involve having an LM play the role of an evaluator and conducting a similar analysis for its judgements.
3. The crowdworkes were asked to rate LM responses according to each criterion in series and then to give an overall score. I wonder such ordering of the task is not itself a confounder. Would overall scores be different if the crowsworkers were not asked to give scores for criteria before? Would a score for `Formatting` be different if the crowdworkers were not ask for `Factuality` before? Do authors control for this anyhow?
4. It’s subtly incorrect to say that “Perez et al. (2023) identified similar ‘inverse-scaling’ behaviour, where training on a RLHF objective worsens sycophancy”. I think the paper showed that base LMs are already pretty sycophantic and that RLHF does not fix that. I think from that paper alone it’s still unclear that RLHF increases sycophancy; there is no robust trend. However, Sharma et al. [3] have recently provided much stronger evidence for the role of human feedback and RLHF objective in the emergence of sycophancy.

[1] https://arxiv.org/abs/2211.03540

[2] https://arxiv.org/abs/2212.08073

[3] https://arxiv.org/abs/2310.13548

**Questions:**

This is not really a flaw, but it would be much better if the experiments were done on responses from frontier models (such as GPT-4, Claude 2 and possibly LLaMA-2-70B). Was budget the limiting factor?

A minor remark: The idea of predicting human preference judgements using a logistic regression model on top of human-interpretable features was recently explored in two concurrent papers that might be worth taking a look at: https://arxiv.org/abs/2310.13548 and https://arxiv.org/abs/2310.13011

---

> ### Author Response · Authors · 2023-11-15
>
> >One big question not addressed in the paper is how the biases in human judgement scale with human capabilities (e.g. question difficulty, annotator competence) and model capabilities (e.g. model size). Are humans more biased when evaluating responses they’re less competent to evaluate (e.g. medical advice, science explanations)? Are larger models more susceptible to exploit human biases? Overall, I think a good perspective for grounding this paper is scalable oversight [1]: the problem of evaluating agents more capable than the evaluators. LM capabilities will soon surpass human capabilities on many tasks and it’s critical to know whether human feedback will be increasingly misleading. That’s of course a big open problem and I don’t take not addressing it against the paper, it but it would be relatively easy to do experiments with models of the same family but different size or group questions by (perceived) difficulty.
>
> This is a really interesting question! Our contribution in this paper is to carefully confirm that the effect exists at all. It would be great to see future work investigate the dynamics of this effect, and how it changes with task, model, problem domain etc.
>
> >I’m not sure I’m convinced that “Human evaluation is necessary” (as stated in the conclusion). Techniques such as RL from AI feedback have been shown to be as effective as RLHF for increasing the harmlessness of LMs [2]. They ultimately rely on humans, but these are human experts involved in prompt engineering and designing rules and constitutions used for generating training data for SFT and training preference models. A big open questions is whether LMs-as-evaluators inherit human biases, even when carefully prompted. A simple experiment addressing this question would involve having an LM play the role of an evaluator and conducting a similar analysis for its judgements.
>
> It's true that "human evaluation with *crowdworkers*" is not always necesary, but language is an inherently human construct and so we would argue that humans are likely to be required to some degree for the foreseeable future. Using models for evaluation is more scalable but these models were still trained on data (ie language) that was originally generated by humans. Further, more targeted experiments would be valuable to confirm whether using LLMs as evaluators mitigates or exacerbates the biases we identify in this paper.
>
> >The crowdworkes were asked to rate LM responses according to each criterion in series and then to give an overall score. I wonder such ordering of the task is not itself a confounder. Would overall scores be different if the crowsworkers were not asked to give scores for criteria before? Would a score for Formatting be different if the crowdworkers were not ask for Factuality before? Do authors control for this anyhow?
>
> Yes, we accounted for this - see the first paragraph in Sec 2.1. We used one groups of annotators to collect overall preference scores, and a separate group for the error annotations (who were also asked for overall ratings). You are correct that there is a 'priming' effect: Fig. 7 in the appendix shows a higher correlation between error scores and overall scores for the group that performed the error annotation, indicating that this did indeed affect their decision. The main results in the paper are based on the 'unprimed' annotations from the separate group, but we consider this an auxiliary finding of interest.
>
> > It’s subtly incorrect to say that “Perez et al. (2023) identified similar ‘inverse-scaling’ behaviour, where training on a RLHF objective worsens sycophancy”.
>
> We have discussed this and agree with your observation: we have updated our paper.
>
> > QA: This is not really a flaw, but it would be much better if the experiments were done on responses from frontier models (such as GPT-4, Claude 2 and possibly LLaMA-2-70B). Was budget the limiting factor?
>
> Our experiments were run using the most performant models available with permissive licenses at the time the experiments were run. GPT-4 and Claude were not available to us for legal reasons.
>
> > QB: A minor remark: The idea of predicting human preference judgements using a logistic regression model on top of human-interpretable features was recently explored in two concurrent papers that might be worth taking a look at: https://arxiv.org/abs/2310.13548 and https://arxiv.org/abs/2310.13011
>
> Thank you! These were both released after the ICLR submission deadline, but we will add them to our related work. We feel that the existence of related contemporary work serves to highlight the importance of this line of research and its growing interest in the community.

---

> > ### Comment · Reviewer_mWEt · 2023-11-21
> > **Thanks for the response**
> >
> > Thanks for the response. You addressed my main concerns and, upon rethinking, I decided to increase my score to 8.
> >
> > > It's true that "human evaluation with crowdworkers" is not always necesary, but language is an inherently human construct and so we would argue that humans are likely to be required to some degree for the foreseeable future. Using models for evaluation is more scalable but these models were still trained on data (ie language) that was originally generated by humans.
> >
> > True, but you can think of language as something needed to bootstrap self-improvement of LLM agents. It's an empirical question whether it's possible. That's a tangent, of course.
> >
> > > Yes, we accounted for this - see the first paragraph in Sec 2.1. We used one groups of annotators to collect overall preference scores, and a separate group for the error annotations (who were also asked for overall ratings). You are correct that there is a 'priming' effect: Fig. 7 in the appendix shows a higher correlation between error scores and overall scores for the group that performed the error annotation, indicating that this did indeed affect their decision. The main results in the paper are based on the 'unprimed' annotations from the separate group, but we consider this an auxiliary finding of interest.
> >
> > Thanks for explaining this!

---

### Official Review · Reviewer_Rbzk · 2023-10-31

**Soundness:** 3 good
**Presentation:** 3 good
**Contribution:** 2 fair
**Rating:** 6
**Confidence:** 3

**Summary:**

This paper studies what attributes human preferences models over LLM output stend to capture, and which they don’t. Specifically, it considers ten error types: harmfulness, fluency, scope, repetition, refusal, formatting, relevance, factuality, inconsistency, and contradiction, and has human annotators label whether a failure arises in any of these categories, then uses lasso on top of these categories to to predict the annotated quality of an output (measured by different annotators). They find that refusal, formatting, relevance, repetition, and factuality, and inconsistency lead to reductions in overall score, while contradictions, scope, fluency, and harmfulness do not. In a second experiment, the annotators find that low assertiveness tends to lower annotator scores, and raising assertiveness tends to improve scores across subcategories (e.g., factuality, relevance). The paper finally finds that RLHF increases assertiveness, suggesting that the annotator preferences get baked into the objective.

**Strengths:**

* The paper studies an important problem: what attributes do human labelers actually care about in model outputs, and does this match the attributes we'd like
* The paper writing is clear throughout and easy to follow
* Many of the empirical results would likely be interesting to the community at large; I especially liked Figure 5, which shows that simply increasing assertiveness reduces the error rates for many categories (e.g., factuality)

**Weaknesses:**

* The paper feels a bit ad-hoc; the properties tested seemed kind of arbitrarily chosen, but the specific set tested should have a significant impact on the learned lasso weights (e.g., if one feature is more predictive than the rest, the lasso weight would put all of it on that)
* Some aspects of the paper do not engage with prior work. For example, it claims that “human feedback the de facto standard for evaluation” with no citation, and does not engage with the extensive work benchmarking LLMs
* The RLHF increases assertiveness claim is tested indirectly; it compares Llama 2 13B and Command 52B, which have many confounders beyond RLHF.

**Questions:**

* How do the lasso weights change when you optimize on different subsets of the properties?
* How much does the choice of ten properties you label for matter? What would happen if you included different properties?

---

> ### Author Response · Authors · 2023-11-15
>
> >The paper feels a bit ad-hoc; the properties tested seemed kind of arbitrarily chosen, but the specific set tested should have a significant impact on the learned lasso weights (e.g., if one feature is more predictive than the rest, the lasso weight would put all of it on that)
>
> We disagree that Lasso will choose a single feature, since the features are not all perfectly correlated. Lasso will include any features that lead to a large enough predictive improvement to counteract the penalty term. However, we have updated the paper with more details of how the error types were chosen.
>
> >Some aspects of the paper do not engage with prior work. For example, it claims that “human feedback the de facto standard for evaluation” with no citation, and does not engage with the extensive work benchmarking LLMs
>
> Thanks for pointing this out - you are correct that we should have included references to LLM benchmarking. We have updated the paper.
>
> >The RLHF increases assertiveness claim is tested indirectly; it compares Llama 2 13B and Command 52B, which have many confounders beyond RLHF.
>
>
>
> Our results do show that Llama2 exhibits higher assertiveness at equivalent quality, but it's true that we cannot be certain *why* this is the case. Since the major difference between the two models is the RLHF training, we suggest that this may be the cause of the difference.
>
> One of our primary contributions is to clearly demonstrate the biasing effect of assertiveness on perceived quality. To the best of our knowledge, this is the first work that demonstrates this for LLMs and we hope this research will provide a basis for future efforts to build on.
>
> > QA: How do the lasso weights change when you optimize on different subsets of the properties?
>
> We have added a sensitivity analysis of the Lasso regression to the paper, see Fig 9 in the appendix. The changes in regression weighting when we leave out each feature in turn are fairly small, and the features that get selected do not change, indicating that the regression is stable. We hope that this addresses your concern.
>
> > QB: How much does the choice of ten properties you label for matter? What would happen if you included different properties?
>
> I'm not sure I fully understand the question - clearly, a different set of properties would lead to different results. However, our intention was to establish a set of error types with as much coverage as possible. We welcome any suggestions for errors that have been ommitted.

---

> > ### Comment · Reviewer_Rbzk · 2023-11-17
> >
> > Thanks for your rebuttal, and especially for adding the Lasso regression analysis! I'm more convinced in the results, and increased my score (although I still think deleting more than one property, or adding new sets of properties could lead to larger changes).
> >
> > I'm still quite worried about the comparison between Llama 2 13B and Command 52B; there are many major differences in the two models (of which RLHF is one). I think it would help the paper to qualify this more clearly, although I don't think it's core to the message.

---

> > > ### Author Response · Authors · 2023-11-20
> > >
> > > Thanks for your response and for increasing your score!
> > >
> > > We did update the paper to be more explicit regarding the limitations of the claims regarding Llama and Command, it was not our intention to claim that this was due to the RLHF. As you rightly point out, we are trying to convey that RLHF may be one of the contributors affecting this observation, but there could of course be others that are not possible to isolate and test for. Thank you for your feedback and for helping to improve the quality and messaging of our work!

---

### Official Review · Reviewer_iLy7 · 2023-10-31

**Soundness:** 3 good
**Presentation:** 3 good
**Contribution:** 3 good
**Rating:** 6
**Confidence:** 4

**Summary:**

This paper analyzes potential confounding factors in human judgment of LLM-generated text, and hypothesizes that these confounders lead to optimizing the wrong metrics in RLHF (i.e., optimizing for assertiveness rather than factuality). The main result is that they find human annotators rely on features such as assertiveness and complexity when judging the overall quality of a piece of text, rather than relying on harder-to-evaluate features like factuality. The paper also evaluates quality judgments for existing models trained with RLHF on human preferences, again finding a correlation between assertiveness and quality judgments.

**Strengths:**

This paper studies a very important problem: that without understanding the underlying factors influencing human judgments of text, in light of ambiguous and underspecified annotation guidelines asking for nebulous quality ratings, when performing standard RLHF we are optimizing models in unintentional directions. The analysis of annotations is relatively thorough and easy to understand.

**Weaknesses:**

A couple of points:
* I wish there were discussion on not just the problem of individual judgments being ambiguous, but that standard RLHF is optimizing towards a single user preference per example rather than considering factors resulting in a distribution over judgments for a number of different annotators (a point that is made e.g., as a motivation for jury learning, Gordon et al. 2022).
* I would also like more discussion on what to do in light of these findings. There were hints at one possible perspective of a solution -- optimizing nebulous "preference" is far from what people do when using language, which is more based on utility and communicative success (which preference is not tied to at all).
* I think the analysis in Section 4 is conflating a few different factors in comparing models -- the fact that Command is trained off-policy and Llama 2 is on-policy isn't discussed in enough depth for me to be convinced this is relevant to mention. I don't have a good intuition on why that might influence the model behavior, especially if the two models are trained on different data as well.
* Following the above, I would have liked to see an experiment that directly shows how optimizing for human preference influences outputs across different dimensions explored in this paper (e.g, assertivness). The evidence from this paper is post-hoc analysis of existing models (in Section 4), and the earlier experiments showing annotators generally conflate quality with features like assertiveness/complexity. There are a couple of things this analysis conflates itself, e.g., the actual data that Command and Llama 2 are trained on, and their learning algorithms. But an even stronger experiment would perform RLHF-based finetuning (or some kind of finetuning with human preference judgments) on data where we have judgments of or could control complexity and assertiveness, and measure how the final models perform on factuality judgments after this process. E.g., perhaps this would show that optimizing for assertiveness++ responses hurts factuality. Or maybe it would show that with the exact same data, fine-tuning using a reward model exacerbates this problem more than just fine-tuning directly. But with the current analysis presented, it's hard to make any concrete conclusion beyond that all these models result in judgments conflating quality with style.
* I would also like to see an experiment controlling for factuality, like in Section 3.2. E.g., if it's possible to even post-hoc separate the results in Figs 3 and 4 by whether an expert judged the output as factual or not. Basically, do you find that assertive false responses are rated as factual by annotators more often than inassertive true responses?
* I think the main table showing expert judgments for Sec 3.2 is in the appendix, but in my opinion it should be in the main paper.

In general, I think the direction this paper investigates is super important, and it indeed highlights a number of issues with current methods. But I also think it could be made a lot stronger, especially in directly showing how these annotation artifacts influence model behavior.

**Questions:**

* Why were the error types in Section 2 the ones chosen? There is a brief discussion (inspiration from Xu et al. 2023, Gricean maxims) but it would be nice to have more discussion on why these particular types were chosen.
* Why do contradiction/scope/fluency/harmful have no contribution in Figure 1?
* Did you analyze where disagreements came from in factuality judgments in Section 2.1? Where agreement was 0.64
* What was the hypothesis of assertiveness and complexity being confounders founded upon? Are there other confounding factors that could influence this, e.g., length?
* "incorrectly perceived as being less repetitive" -- is repetitiveness objective? It seems a lot more subjective than something like factuality.
* What are the two lines in Figure 6?
* Figure 5 shows annotated error rates -- are these expert annotations or general crowdsourced annotations?

---

> ### Author Response · Authors · 2023-11-15
> **Response to weaknesses**
>
> >I wish there were discussion on not just the problem of individual judgments being ambiguous, but that standard RLHF is optimizing towards a single user preference per example rather than considering factors resulting in a distribution over judgments for a number of different annotators (a point that is made e.g., as a motivation for jury learning, Gordon et al. 2022).
>
> Thanks for the pointer to the jury learning paper, we have added this to our discussion. We briefly mention some related work on modelling annotator distributions by [Wang et al., 2023](https://direct.mit.edu/tacl/article/doi/10.1162/tacl_a_00584/117217). Collecting multiple annotations will partially mitigate the problem, but care still needs to be taken when aggregating to account for the biases that we present here. There are many [issues with RLHF](https://arxiv.org/abs/2307.15217) but our intention was to perform a targeted analysis of whether preference scores are a good training/evaluation metric.
>
> >I would also like more discussion on what to do in light of these findings. There were hints at one possible perspective of a solution -- optimizing nebulous "preference" is far from what people do when using language, which is more based on utility and communicative success (which preference is not tied to at all).
>
> Thank you, this is a very valid point! Testing possible mitigation strategies is outside the scope of this paper, but we have added more detailed suggestions for possible future directions.
>
> >I think the analysis in Section 4 is conflating a few different factors in comparing models -- the fact that Command is trained off-policy and Llama 2 is on-policy isn't discussed in enough depth for me to be convinced this is relevant to mention. I don't have a good intuition on why that might influence the model behavior, especially if the two models are trained on different data as well.
> >Following the above, I would have liked to see an experiment that directly shows how optimizing for human preference influences outputs across different dimensions explored in this paper (e.g, assertivness). [...]
>
> We are the first to demonstrate the existence of a biasing effect due to assertiveness, which is not a trivial finding. We believe it is unfair to critique the paper for not also including an additional large-scale experiment investigating the effects of RLHF.
>
> We disagree that the only conclusion is that "all the models conflate quality and style": the results definitely show that Llama 2 exhibits higher assertiveness at equivalent quality. However, it is a valid criticism that we cannot be sure *why* this is. We tried to qualify the findings in Section 4 as "preliminary evidence" since we did not directly test the same models with and without RLHF. A targeted set of experiments, using the same base model and varying the number of RLHF steps (and other factors), is planned for the future but exceeds the scope of this paper.
>
> > I would also like to see an experiment controlling for factuality, like in Section 3.2. E.g., if it's possible to even post-hoc separate the results in Figs 3 and 4 by whether an expert judged the output as factual or not. Basically, do you find that assertive false responses are rated as factual by annotators more often than inassertive true responses?
>
> Here are the annotator factuality error rates, grouped by preamble type and expert factuality judgement:
>
> | Preamble Type   |   Factual  |   Not Factual |
> |:----------------|----------------------------:|----------------------------:|
> | Assertiveness\-\-  |                     1.88679 |                     0       |
> | Normal          |                     1.04 |                    16.66  |
> | Assertiveness++ |                     2.17 |                     3.57 |
>
>
> So, it's not the case that "assertive false responses are rated as factual by annotators more often than inassertive true responses", but the factual errors are much less likely to be spotted in assertive outputs. Note that there are only 7 examples that are inassertive and false, so the statistical power of this analysis is weak, and we therefore will not include it in the paper. Expert annotations are unfortunately difficult to scale!
>
>
> >I think the main table showing expert judgments for Sec 3.2 is in the appendix, but in my opinion it should be in the main paper.
>
> We would like to include the full results in the main flow of the paper, but unfortunately space is limited. We have the choice to either show annotator and expert judgements alongside each other and require the reader to take the difference, or to show just the difference and include the breakdown of results in the appendix. Presenting the results as differences demonstrates most clearly the findings we are trying to present, without excess complication, but we include the full detailed results in the appendix for completeness.

---

> > ### Author Response · Authors · 2023-11-15
> > **Response to questions**
> >
> > > QA: Why were the error types in Section 2 the ones chosen? There is a brief discussion (inspiration from Xu et al. 2023, Gricean maxims) but it would be nice to have more discussion on why these particular types were chosen.
> >
> > Thanks for the useful feedback! We have updated the paper to include more detail on how the error types were selected.
> >
> > > QB: Why do contradiction/scope/fluency/harmful have no contribution in Figure 1?
> >
> > We address this at the top of Page 4 - scope/fluency/harmful errors were very rare, while contradiction seemingly doesn't have a strong enough influence on overall rating to be selected by the Lasso model.
> >
> > > QC: Did you analyze where disagreements came from in factuality judgments in Section 2.1? Where agreement was 0.64
> >
> > We did not attempt to categorise the types of disagreement. However, based on our experience from performing the expert annotations ourselves, some statements in outputs required significant effort to verify (e.g., checking a software user manual to confirm whether a menu item exists or not), and other statements were plausible but wrong.
> >
> > > QD: What was the hypothesis of assertiveness and complexity being confounders founded upon? Are there other confounding factors that could influence this, e.g., length?
> >
> > Yes, other confounding factors are likely to exist! Length does have an effect on scores, we include an analysis in Fig 8 in the appendix. As we mention in the text, this is complicated by the fact that long answers are more repetitive (and therefore more annoying) and short answers simply contain fewer facts that could be incorrect. We selected assertiveness and complexity as they are intuitive yet undesirable properties of LLM outputs, but there could be other factors worth investigating in future.
> >
> > > QE: "incorrectly perceived as being less repetitive" -- is repetitiveness objective? It seems a lot more subjective than something like factuality.
> >
> > You are correct, repetition is not an entirely objective property. However, the instructions refer to repetition of phrases or items in a list, so it is more tightly defined in the context of our experiments.
> >
> > > QF: What are the two lines in Figure 6?
> >
> > Thanks for the feedback - these are the trendlines for Command 52 and Llama 2. We have updated the caption accordingly.
> >
> > > QG: Figure 5 shows annotated error rates -- are these expert annotations or general crowdsourced annotations?
> >
> > These are crowdsourced annotations - we have updated the caption accordingly.

---

> > ### Comment · Reviewer_iLy7 · 2023-11-22
> >
> > Thanks for the response!
> >
> > I don't mean to ask for additional experiments -- just refinement of how the results/conclusions are presented in Section 4. With these experiments, I don't think we can conclude that it's solely RLHF influencing differences in assertiveness. I think I am mainly confused because the distinction between on vs. off-policy and RLHF vs. non-RLHF is not very precise. There are a couple elements here: whether we are using human judgments for training in any way, whether we're training a reward model and performing online RL with it. One could perform online RL to train/finetune an LM without any human feedback (e.g., reward models being BLEU scores, see some of Julia Kreutzer's work on machine translation from a few years ago), and one could train with human feedback without on-policy learning (essentially imitation learning with loss coefficients derived from human feedback). Or the other combinations: RL with a reward model trained on human feedback, standard imitation learning without human feedback (i.e. the standard language modeling objective)

---

> > > ### Author Response · Authors · 2023-11-22
> > >
> > > Thanks for following up!
> > >
> > > We did update the paper to be more explicit regarding the limitations of the claims in Section 4; it was not our intention to claim that this was definitely due to the RLHF. As you rightly point out, we are trying to convey that RLHF may be _one_ of the contributors affecting this observation, but there could of course be others that are not possible to isolate and test for. Thank you for your feedback and for helping to improve the quality and messaging of our work!

---

### Official Review · Reviewer_knEY · 2023-11-04

**Soundness:** 3 good
**Presentation:** 3 good
**Contribution:** 3 good
**Rating:** 6
**Confidence:** 4

**Summary:**

This paper studies an important topic that is very prevalent these days in the LLM field especially (but generally as well)  - use of human feedback to evaluate large language models as well as use the coarse grained data of human preference to train/tune the LLMs - the work studies, defines, disentangles and subjectivity of the human preferences and tries to ‘explain’ how various types of errors are correlated to this preference. It further suggests that human preferences do not correlate well to the important dimension of factuality of the LLM generated information.

In particular, the paper tries to define a broad set of discrete dimensions of various properties/attributes of fine-grained evaluation criteria for LLM model for human judgment - this is trying to map the one preference score to the underlying error categories that the one score should ideally represent (somewhat equally in principle) - however these categories are meant to be defined to capture the epistemological meaning of the human preference and map that to a simple oenology of the error types described in the paper.

The paper then uses 3 datasets and collects the human preference annotations (with quality control and best practices) )on the various recent model outputs and the ontological error types. The  paper uses a Lasso regression model to fit the preference score to error categories for studying how various error categories contribute to the overall mapping to the single human preference. Some error types have a stronger contribution to the mapping function (eg. Refusal to answer which is very objective dimension unlike others) which some other do not contribute (eg. Harmfulness). Some other important dimensions like factuality and inconsistency have less weights compared to the others.

The paper then tries to study the ‘halo effect’ cognitive bias of the LLM generated output and how humans perceive and prefer more assertive generation and more complex generation over less assertiveness and less complexity. The paper notes that increased assertiveness and complexity both lead to slightly higher perceived quality, while low assertiveness
leads to the worst rated responses. respond. Authors also note that non-expert annotators tend to underestimate the rate of inconsistency or factuality errors, and they are less likely to spot these errors in outputs that are assertive. The authors suggest how controlling the prompts for assertiveness (eg. having low assertiveness) could perhaps be used to promote safer refusals.

Since perceived preference is correlated with assertiveness, the relationship suggests that using human feedback as a training objective could inadvertently increase the complexity and assertiveness of outputs (as a side-effect) and thus introduce these biases in the trained models.

**Strengths:**

- RLHF and human preference alignment topic for LLM human-AI alignment is one of the highly discussed topics today for chatbot training. RLHF is used to teach the models to generate safer outputs (for example, refuse to respond when inappropriate, promote politeness and reduce toxicity and biases). The studies in the paper suggest that the human preferences can easily get confounded with assertive and complex text and prefer those therefore introduce this 'assertiveness bias' and 'complexity bias' in the modeling process.

- The paper also discusses important findings around factuality (and the important topic of hallucinations studies in LLMs). "Crowdworkers underestimate the rate of factuality and inconsistency errors. This difference is increased for high assertiveness responses, and decreased
for low assertiveness responses. In other words, annotators are more trusting of assertive responses, and are less likely to identify factuality or inconsistency errors within them" -> in other works it could be possible that RLHF methods may promote hallucinations, and introduce more assertiveness and complexity in the generated output.

**Weaknesses:**

- it is not clear if this error type categorization is comprehensive enough; authors haven't provided any empirical/experimental support for these error categories.
- There are other important biases from AI safety perspective that isn't explicitly studied in the paper (could possibly make the paper more relevant) - hallucinations, more fine-grained categorization of inconsistencies, toxicity, etc
- the datasets studied may not be comprehensive enough to have captured some of the important dimensions of AI safety (toxicity, attribution, etc)
- Besides the Lasso regression model, other interpretable/explainable ML methods might have shed more light on error categories and their correlation with both human preference and model output.
- authors note that human preferences are mostly absolute or relative - there isn't any comparison between the two in this research. Since the paper provides insights into understanding these human preferences, more focus on various types of human preferences would have strengthened the finding of the paper
- Authors very briefly touch upon the AI safety implications but do not give justice to this very important piece of information in the paper - what are the AI safety implications as a result of using human preferences for training and evaluation.

**Questions:**

(please address the weaknesses above)

---

> ### Author Response · Authors · 2023-11-15
>
> > it is not clear if this error type categorization is comprehensive enough; authors haven't provided any empirical/experimental support for these error categories.
>
> Thank you for the feedback! The error categories were based partly on the well-established Gricean Maxims (e.g. the Maxim of Quantity prohibits repetition), partly on experimental work by [Xu et al.](https://aclanthology.org/2023.acl-long.181/) who asked human evaluators to give justifications for their scores, and influenced by the factors that users care about when using LLMs in production environments. We did our best to account for as wide a range of errors as we could, but it's possible that we have missed something. We have updated the paper with more details.
>
> > the datasets studied may not be comprehensive enough to have captured some of the important dimensions of AI safety (toxicity, attribution, etc)
>
> We selected datasets that are well-scoped (such that evaluation is possible) but are still challenging and require long-form generation. Further work is definitely needed - but there is a lack of datasets with permissive licenses for realistic use cases.
>
> > Besides the Lasso regression model, other interpretable/explainable ML methods might have shed more light on error categories and their correlation with both human preference and model output.
>
> Thanks for the feedback - we have released the full dataset and welcome future work investigating more detailed interactions between error types and preferences.
>
> >authors note that human preferences are mostly absolute or relative - there isn't any comparison between the two in this research. Since the paper provides insights into understanding these human preferences, more focus on various types of human preferences would have strengthened the finding of the paper
>
> Time and budget constraints meant that we used a fixed protocol, based on RankME (Novikova et al., 2018), and focussed on identifying whether the biasing effects exist at all. You're right that it would be really interesting to run our experiments with different protocols: varying the number of outputs evaluated at a time, using absolute/relative scores, different instructions etc. Future work could investigate whether the strength of the effect varies with annotation protocol.
>
> > There are other important biases from AI safety perspective that isn't explicitly studied in the paper (could possibly make the paper more relevant) - hallucinations, more fine-grained categorization of inconsistencies, toxicity, etc
> >Authors very briefly touch upon the AI safety implications but do not give justice to this very important piece of information in the paper - what are the AI safety implications as a result of using human preferences for training and evaluation.
>
> Safety is an important area of research, and we agree that there would be a lot of value in extending our study to consider the interaction between assertiveness/complexity and various types of unsafe LLM behaviour. However, we believe that doing this justice would be another whole paper in itself.
>
> In this paper we confirm that preference scores are a) an imperfect measure of important error types, and b) biased by assertiveness. These observations hold true even for a (hypothetical) perfectly safe model. We hope it will serve as a starting point for further work, including experiments specifically targeted at the wide range of important safety concerns.

---

### Meta-Review · Area_Chair_NdRg · 2023-12-02

**Metareview:**

This paper explores the nature of human feedback and preference scores used in LLM training.  An experiment is done where annotators label outputs from LLMs according to a number of criteria.  Based on understanding how overall preferences are predicted from individual criteria, the paper finds that factuality and inconsistency are less predictive of overall scores than they should be.  In a controlled experiment by varying prompts, making models more assertive can bias factuality judgments. Given that perceived quality is correlated with assertiveness, RLHF may drive models to be more assertive in a way that is ultimately not aligned with downstream goals.

The reviewers found the topic of this paper very timely and generally found it well-written. In my view as AC, the human study conducted here is quite valuable and will be useful to others in the community.

There were a number of critiques of this paper. knEY and Rbzk bring up a question of comprehensiveness of the error categories. I see this critique, although from the discussion, I don't think any concrete directions for improvement emerged, and in general I don't think there's ever a perfect taxonomy of errors.  iLy7 brings up several ways the paper could be stronger, such as by recognizing that different annotators may have different preferences or by emphasizing what can be done given the findings.  mWEt mentions possible ramifications of model sizes and whether it's possible to understand human biases further.

I would strongly encourage the authors to revise their paper to try to address these points. However, I recognize that not all of these follow-up studies are addressable within the constraints of a 9-page ICLR paper.

**Justification For Why Not Higher Score:**

This paper presents a useful study but isn't the "last word" on this topic. It could be more comprehensive and more could be done in these directions.

**Justification For Why Not Lower Score:**

In general, this paper presents useful analyses on a timely topic. This is likely to have an impact on the conversation around human preference judgments and RLHF.

---

### Decision · Program_Chairs · 2024-01-16

Accept (poster)